# A mixed methods survey of social anxiety, anxiety, depression and wig use in alopecia

Kerry Montgomery,[1] Caroline White,[2] Andrew Thompson[1]

► Prepublication history and additional material are available. To view please visit the journal (http://dx.doi.org/10.1136/bmjopen-2016-015468).

[1]Department of Psychology, University of Sheffield, Sheffield, UK
[2]Dermatopharmacology Unit, University of Manchester, Manchester, Greater Manchester, UK

**Correspondence to**
Kerry Montgomery;
kmontgomery1@sheffield.ac.uk

## ABSTRACT

**Objectives** This study aimed to examine levels of social anxiety, anxiety and depression reported by people with alopecia as a result of a dermatological condition and associations with wig use. The study also sought to report on experiences of wearing wigs in social situations and the relationship with social confidence.

**Design** A cross-sectional survey was sent by email to the Alopecia UK charity mailing list and advertised on social media.

**Participants** Inclusion criteria were a diagnosis of alopecia, aged 13 or above and sufficient English to complete the survey. Exclusion criteria included experiencing hair loss as a result of chemotherapy treatment or psychological disorder. Participants (n=338) were predominantly female (97.3%), Caucasian (93.5%) and aged between 35 and 54 years (49.4%) with a diagnosis of alopecia areata (82.6%).

**Main outcome measures** The Social Phobia Inventory measured symptoms of social anxiety, and the Hospital Anxiety and Depression Scale was used to measure symptoms of anxiety and depression. Survey questions were designed to measure the use of wigs. Open-ended questions enabled participants to comment on their experiences of wearing wigs.

**Results** Clinically significant levels of social anxiety (47.5%), anxiety (35.5%) and depression (29%) were reported. Participants who reported worries about not wearing a wig reported significantly higher levels of depression: t(103)=3.40, p≤0.001; anxiety: t(109)=4.80, p≤0.001; and social anxiety: t(294)=3.89, p≤0.001. Wearing wigs was reported as increasing social confidence; however, the concealment it afforded was also reported as both reducing fear of negative evaluation and maintaining anxiety.

**Discussion** Overall, 46% of participants reported that wearing a wig had a positive impact on their everyday life with negative experiences related to fears of the wig being noticed. Psychological interventions alongside wig provision would be beneficial for people living with alopecia.

## INTRODUCTION

For many people hair is a central aspect of appearance and self-image; therefore, hair loss can have a negative impact on self-esteem, body image and confidence.[1 2] While hair loss may have few physical health

### Strengths and limitations of this study

► This is the first study to examine National Health Service wig provision and how people living with alopecia finance wigs.
► Validated measures of social anxiety, anxiety and depression were used to determine levels of distress.
► Survey questions investigated the experience of wearing wigs.
► The sample was predominantly female and Caucasian; therefore, further research is needed to examine experiences of men and cultural differences in alopecia.
► Participants recruited from Alopecia UK may have been experiencing higher levels of distress than the general population, having accessed support by the charity.

consequences, living with a condition that affects appearance can have a significant impact on everyday functioning.[3–6] Studies have shown that people living with alopecia are at a higher risk of developing depression, anxiety and social phobia than the general population.[7 8]

Higher levels of body dissatisfaction and preoccupation with appearance have been reported in men and women living with alopecia, relative to controls,[7 9 10] with 40% of women reporting marital problems as a consequence.[3] Perception of hair loss has a significant impact on quality of life, accounting for 35% of the variance.[11]

Wearing a wig to conceal hair loss is an important coping strategy for many individuals[12–15] giving people the confidence to return to work and engage in social activity.[14] We have known since the time of Goffman's seminal work on stigmatisation[16] that people may choose to conceal attributes they believe will be discredited by others. Recent studies indicate that alopecia areata can be associated with higher levels of perceived stigmatisation than concealable conditions such as mental health conditions.[17] Wearing a wig might reduce an individuals' perception of a

discrepancy between cultural norms regarding the appearance of the hair and their own appearance. However, wearing a wig might not be an ideal solution as the fear of being 'discredited' should the wig be noticed remains.[16]

Despite wigs being an important coping strategy for people living with alopecia, many people experience difficulties accessing wig prescriptions via the National Health Service (NHS). Some NHS trusts report that alopecia is a 'cosmetic issue'; therefore, the financial burden of wigs falls with the patient. Further cuts in dermatology could mean more patients have difficulties accessing wigs, which may have significant psychosocial consequences for people living with alopecia. Access to psychological support within dermatology is limited,[18] and currently there are no psychological interventions available to target the specific needs of this population.[19] The need for psychological interventions for alopecia has been emphasised by patients and clinicians.[20]

The current mixed methods study aims to examine (1) levels of social anxiety, anxiety and depression reported by people living with alopecia, (2) associations between wig behaviours and psychosocial distress and (3) experiences of wearing wigs in social situations. The current study will also investigate how participants access wigs and the financial implications of wearing wigs.

## Method

A cross-sectional survey incorporating validated measures of social anxiety, depression and anxiety, questions on wig usage, financing wigs, and open-ended questions regarding experiences of wearing wigs in social situations was sent by email to the Alopecia UK mailing list and advertised on social media. Ethical approval for the study was gained from the University of Sheffield Psychology Department Ethics Committee. Inclusion criteria were: a diagnosis of alopecia as a result of a dermatological condition, aged 13 or above and sufficient English proficiency to complete the survey. Participants aged under 16 years of age were advised to complete the questionnaire with a parent or guardian. Exclusion criteria included experiencing hair loss as a result of chemotherapy treatment or primary psychological disorder (eg, trichotillomania).

## Measures
### Social anxiety
The Social Phobia Inventory (SPIN)[21] is a 17-item self-report questionnaire to measure various elements of social anxiety (fear in social situations, avoidance of social situations/performances and physiological symptoms in social situations). Connor et al[21] reported a mean score of 41 on the SPIN for individuals with social phobia, compared with a score of 12 in a non-psychiatric population. A score of 19 or above indicates symptoms of social anxiety with 79% diagnostic accuracy.[21]

### Depression
The nine-item self-report Patient Health Questionnaire[22] was used to measure symptoms of depression experienced over the last 2 weeks. A mean score of 3 has been reported by healthy controls, in comparison with a score of 17 in people diagnosed with depression.[22] A cut-off score of ≥10 was used to identify individuals experiencing depression (88% specificity and 88% sensitivity).[23]

### Anxiety
The Generalised Anxiety Disorder Questionnaire[24] is a seven-item self-report scale to measure generalised anxiety symptoms over the last 2 weeks. A mean score of 4.9 has been reported in healthy controls, in comparison with a score of 14 in people living with generalised anxiety.[24] A cut-off score of ≥8 was used to identify individuals experiencing anxiety.[25]

### Survey questions
A number of survey questions were included in which participants were asked to report on the situations in which they would choose to wear a wig (eg, social situations, dating, work/university/school) and if they had any worries about financing or not wearing wigs. Open-ended questions were asked to elicit experiences of wearing wigs in social situations. Questions included: Has wearing a wig had any impact on your confidence during social situations?, Do you have any worries about not wearing a wig? and How has wearing a wig affected your everyday life?

### Analysis
Statistical analysis was carried out using SPSS V.22.[26] Descriptive statistics were used to provide information on (1) demographic and clinical variables, (2) percentage of people wearing wigs and the situations in which people would choose to wear wigs, (3) wig funding and (4) clinically significant levels of social anxiety, depression and anxiety. Associations were tested between demographic variables, wig behaviour (eg, situations in which people would choose to wear wigs) and social anxiety, depression and anxiety using independent samples t-tests and Pearson correlation as appropriate. An alpha level of 0.05 was used for all statistical tests.

Qualitative responses were analysed using qualitative content analysis.[27] Qualitative Content Analysis is a systematic text analysis technique that preserves the advantages of quantitative analysis, allowing frequencies of data to be reported while also providing a method of interpreting participant experiences of wearing wigs during social situations.[27]

An inductive approach to data analysis was used to determine emergent themes generated through examination of participant's responses to specific questions. The data were analysed based on the wording used by participants to describe their experiences and organised into higher order themes. For example, the response that wigs 'improved confidence' and participants 'felt more able to go out' was coded into the themes 'improved confidence' and 'positive experience of wearing a wig'. Frequency of the occurrence of themes across participants was recorded by the first and second authors. Inter-rater reliability

**Table 1** Participant characteristics

|  | n | % |
|---|---|---|
| **Type of alopecia** |  |  |
| Alopecia areata | 114 | 33.7 |
| Universalis | 106 | 31.4 |
| Totalis | 59 | 17.5 |
| Frontal fibrosing | 7 | 2.1 |
| Androgenic | 8 | 2.4 |
| Male pattern baldness | 3 | 0.9 |
| Scarring | 3 | 0.9 |
| Female pattern baldness | 2 | 0.6 |
| Lichen planopilaris | 1 | 0.3 |
| Folliculitis decalvans | 1 | 0.3 |
| Chemically induced | 1 | 0.3 |
| Unknown | 32 | 9.5 |
| **Gender** |  |  |
| Female | 329 | 97.3 |
| Male | 5 | 1.5 |
| No response | 1 | 0.3 |
| **Ethnicity** |  |  |
| White (Caucasian) | 316 | 93.5 |
| Asian (British Asian) | 10 | 3.0 |
| Black (Black British) | 5 | 1.5 |
| Other | 4 | 1.2 |
| No response | 1 | 0.3 |

(98%) was established by determining any differences in coding between the first and second authors.

## RESULTS
### Participants

Participants (n=338) were predominantly female (97.3%) and Caucasian (93.5%). All participants lived in the UK. The majority of the sample were aged between 35 and 54 years (49.4%) (13–17 years 1.5%, 18–25 years 8%, 26–34 years 13.6%, 55–64 years 18.6% and 65 years and over 8.9%). Eleven different types of alopecia were reported across the sample (table 1), and the majority of the sample comprised of alopecia areata (82.6%). Of the respondents, 10.7% were receiving medical treatment for hair loss including steroid injections, spironolactone and topical treatments.

The majority of participants chose to wear wigs most or all of the time (76%) (table 2). The most common type of wigs worn by participants were acrylic (acrylic monofilament 39.3%, acrylic lace front 27.5% and acrylic wefted wig 14.8%). The majority of participants did not obtain NHS wig prescriptions (50.6%) out of which 11.2% of participants had been told that they were not eligible and 22.5% had never enquired about it. Of the participants who claimed NHS wig prescriptions (46.1%), 11.5%

**Table 2** Wig use of participants (n=338)

| How often do you wear a wig? | % of people wearing a wig |
|---|---|
| Never | 23.6 |
| Occasionally | 16.6 |
| Most of the time | 26 |
| All the time (excluding the night) | 44.1 |
| All the time (including the night) | 5.9 |

claimed one per year, 26% claimed two per year and 8.6% claimed more than two each year.

### Social anxiety, anxiety and depression

Clinically significant levels of anxiety (score ≥8) were reported by 35.5% of participants, with 40.5% reporting no symptoms, 14.5% reporting mild symptoms (score 7–9), 13.5% reporting moderate symptoms and 15% reporting severe symptoms of anxiety. Clinically significant levels of depression (score ≥10) were reported by 29% of the sample, with 43.6% reporting no symptoms, 27.6% reporting mild symptoms (score 5–9), 14.3% reporting moderate symptoms (score 10–14), 8.1% reporting moderately severe symptoms and 6.6% reporting severe symptoms of depression. Clinically significant symptoms of social anxiety (score ≥19) were reported by 47.5% of the sample, with 23.4% reporting mild symptoms, 11.1% reporting moderate symptoms, 7.2% reporting severe symptoms and 6.6% reporting very severe symptoms. Participants reporting worries about not wearing a wig reported significantly higher levels of depression: t(103)=3.40, p≤0.001; anxiety: t(109)=4.80, p≤0.001; and social anxiety: t(294)=3.89, p≤0.001.

### Wig use

Participants were asked when they wear a wig across a number of everyday activities (table 3). The majority of participants reported wearing a wig to socialise (86.7%) and 66.3% of respondents reported they would not feel confident leaving the house without a wig. Of the participants who purchased wigs via NHS prescription, 28.1% reported that they would be unable to afford their wig privately and 65.1% of the sample reported worry about affording new wigs.

**Table 3** Situations in which people would wear a wig (n=338)

| Situations when you would wear a wig | % of people wearing a wig |
|---|---|
| Meeting new people | 78.4 |
| Dating | 61.2 |
| Socialising | 86.7 |
| Work/school/university | 76.3 |
| All the time | 55.9 |

## Qualitative findings

A summary of responses to open-ended survey responses is reported below.

*Has wearing a wig had any impact on confidence during social situations?*

Three hundred and thirteen participants responded to the question on the impact of wigs on confidence in social situations. Of responses, 26% reported a positive impact of wearing a wig. The positive impact was related to the reduced likelihood of comments regarding hair loss in 23% of participants and feeling more confident going out in public (32%).

> Interacting with people who I've never met before, without a wig they may wonder why I have no hair, it is not, the norm to have patchy bald head. Hate the idea people will talk about me, or feel I have a more serious illness.
> At different times in my life a wig has been a survival tool for me.

Participants also reported that wearing a wig could have a negative impact on confidence during social situations (43%). These responses were categorised into two main themes: worries about others knowing it was a wig (47%) and concerns about the wig coming off or discomfort (39%). Wearing a wig also led to reduced activity (41%) in particular sports were avoided due to concerns about having to take off the wig.

> Even 10 years on, I still wonder if people can tell I wear a wig and especially when I meet new people. It has stopped me going to places or trying new activities.

Difficulty adjusting to a change in appearance when wearing a wig was reported with participants describing feeling disconnected from their appearance in a wig.

> No matter how attractive I look in a wig I feel like a sham. I feel the real bald me is not attractive at all.

For some participants (7.8%) wearing a wig had no impact on their confidence, which was due to positive experiences of wearing a wig. Other responses were not coded as participants had simply responded 'no', which was interpreted as 'no impact', but it was unclear if this was related to a positive or negative impact of wigs.

### How has wearing a wig affected your everyday life?

Of the 299 responses, 23% of participants reported that wearing a wig improved their confidence/self-esteem in everyday life.

> It's improved my confidence as I avoided harsh overhead lights in retail stores as it emphasised my thinning hair particularly distressing when I saw my reflection in a mirror. Now I can walk past a mirror with harsh lighting overhead and check my hair and smile!

> It's given me a lot more confidence in just leaving the house and being in front of other people.

Negative views of appearance without a wig were reported (14%).

> Don't like anyone seeing me without it, feel naked.
> I feel a lot less feminine without hair.

Participants reported negative reactions from others if they were not wearing a wig (9.6%), suggesting wigs were used as a coping strategy to manage negative reactions.

> It is an uncomfortable but manageable part of my life. It means that I don't get as many pitying glances from strangers.
> I feel more confident to go out with my children, it becomes incredibly tiring been stared at and shouted at if I ever left the house without my wig on.

Overall, 46.6% of participants reported that wearing a wig had a positive impact on their everyday life, reducing the likelihood of negative reactions while improving confidence with regards to appearance.

Participants also reported problems with wigs that had a negative impact on everyday life (49%; subthemes included: wigs coming off, expense, quality and fitting).

> Wearing a wig is a way of life to me as I've never had hair, but always dreamed of being normal. Not worrying if it's going to fall off or move.
> Every time I step outside my front door I think about weather conditions, who I might meet, if anyone is going to stand too close to me, what they will see, what they will think.

In addition, wearing wigs reduced social activity (33%) in particular; dating, meeting new people and exercise were situations in which participants experienced problems.

> It's depressing every morning having to wake up and put my wig on. When I sleep with someone new in my room I don't take my wig off.
> Constant worry that someone/child will randomly pull the hair, or know it is a wig and make comments. Still anxious when out, but at least I now leave the house.
> Going swimming is difficult I don't like swimming hats, and use an old wig, but can't dive in a pool or swim under water with a wig on, constantly scared of losing wig in water, but love to swim.

Participants reported feeling more self-conscious about having to wear a wig (17%). Self-consciousness appeared to arise as a result of concerns about others finding out they were wearing a wig (22%).

> The NHS wigs look like wigs so I always imagine people are saying 'Look at her wearing that stupid wig'. I only go out if absolutely necessary and avoid public places as often as I can. I FEEL LIKE A FREAK. I've had bad experiences at work and socially where someone took my wig for a bet/ cos they were drunk.

## Worries about not wearing a wig

Participants reported worries about not wearing a wig (40.5%) due to the reactions of others to hair loss (staring 42% and comments 18%).

> People as a rule have hair, why would I want people to stare and comment?

Specific concerns about not wearing a wig included worries others would assume they had cancer (n=13).

> People assume I have cancer. I look strange.

Self-conscious emotions and negative self-appraisals of appearance also led to worries about not wearing a wig (19%).

> I am worried people will see how ugly I really am if they see me bald.

## DISCUSSION

The first aim of the current study was to investigate the prevalence of social anxiety, anxiety and depression in people living with alopecia. Social anxiety was commonly reported by the participants in this study with 47.5% indicating clinically significant levels of social anxiety. While few quantitative studies have examined the prevalence of social anxiety in people living with alopecia specifically, qualitative studies suggest that alopecia can have a significant impact on social functioning.[2–4 13 14] The prevalence of anxiety and depression reported are consistent with previous studies[2] and highlights the importance of identifying coping strategies to help people living with alopecia to manage psychological distress.

The current study also aimed to examine the relationship between psychological variables and wig usage. Worry about not wearing a wig was related to higher levels of depression, anxiety and social anxiety, and with 65% of participants reporting worry about affording new wigs, it suggests the financial burden of affording wigs could lead to additional distress in patients already reporting significant levels of social anxiety, depression and anxiety. This finding has important implications in the current economic climate and suggests changes in NHS wig provision could have negative psychosocial consequences should people with alopecia no longer be able to access wig prescriptions.

For some people living with alopecia, this study clearly shows that wearing a wig is an important coping strategy, helping people to manage negative reactions from others and improving confidence to engage in social activity. However, despite wigs being viewed as an important coping strategy, only 46.7% of the participants who chose to wear wigs accessed NHS wig prescriptions, with some NHS trusts failing to provide any wig provision for patients with alopecia.

The final aim of the current study was to explore participants' experiences of wearing wigs in social situations. The current findings suggest that the relationship between wigs and social confidence is complex, and wigs alone do not appear to reduce social distress for a large number of patients. The current findings indicate that people living with alopecia face a difficult choice in social situations, negative reactions as a result of hair loss being visible or negative reactions as a result of wearing a wig. Managing the noticeability of wigs appeared to lead to significant negative interpersonal consequences, including avoidance of social situations and exercise. Despite generating anxiety and causing discomfort, participants were still choosing to wear wigs suggesting wigs were seen as the 'lesser of two evils' given worries about reactions of others to hair loss. These findings are consistent with previous studies and suggest that wearing a wig does not signify an end point to managing social distress for people living with alopecia.[13 14]

Critical self-appraisals of appearance without a wig ('freak, alien, ugly') appeared to be a contributing factor to wearing a wig, as wigs improved confidence in social situations through reducing appearance concern. Hair loss was viewed as a negative attribute with participants reporting experiences of stigma including comments and staring, and perception of hair loss as a sign of illness. Hair loss is reported as confirmation of identity for cancer patients,[28] which has social implications for people living with alopecia, in that they are often misidentified as 'ill'.

Negative reactions from others represent threats to an individual's desire to be valued and accepted by others.[16] Reactions to social rejection may be prosocial, involving an increased desire to be socially accepted; antisocial, whereby the individual attempts to defend themselves or express anger following rejection; and avoidant,[29] in which an individual may seek to avoid further rejection that could involve reduced social engagement. Wearing a wig may therefore be seen as a strategy to increase social acceptance and to avoid rejection. Indeed, Goffman's seminal work on stigma[16] suggests that individuals may choose to conceal attributes they believe will be discredited by others, in this case hair loss; however, the consequence of this is continued concealment and anxiety about being 'found out'.

There are several limitations of the current study that should be noted. First, the sample is predominantly female and Caucasian; therefore, further research is needed to explore the experiences of men and cultural differences in experiences of alopecia. Second, a number of responses could not be coded into themes as the responses did not relate to the question suggesting some participants may have experienced problems interpreting the questions. Third, participants were recruited from Alopecia UK; therefore, some participants may have been experiencing higher levels of distress than the general population having accessed support via the charity. Finally, given that the participants were recruited from the Alopecia UK mailing list and social media, it is not possible to ascertain a response rate in the current study.

The current findings have important implications when providing support to people living with alopecia and suggest that wigs appear to be much more than

a 'cosmetic' tool and are used to reduce potential experiences of stigmatisation and improve social confidence. Given the significant negative social consequences that could arise should wigs no longer be an option, the current findings should be considered when funding decisions are made regarding wig provision for patients with alopecia. Negative experiences of wearing wigs are related to fears of the wig being noticed and hair loss being discovered; therefore, psychological interventions, particularly those targeting self-conscious emotions such as shame and social anxiety, would be beneficial for people living with alopecia alongside wig provision. To date, there are no studies examining the effectiveness of psychological interventions specific to people living with alopecia; however, approaches such as cognitive–behavioural therapy and compassion focused therapy may be beneficial in targeting processes underlying social anxiety. Attentional training techniques such as mindfulness may also be beneficial to reduce preoccupation with appearance-related thoughts.[30]

## Supplementary data

The authors would like to thank Alopecia UK for their support in recruitment for this study.

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

**Contributors** AT and KM developed the protocol for this study. All authors have contributed to the preparation of this manuscript. KM carried out data analysis for quantitative data, and KM and CW analysed qualitative data to ensure reliability of coding. All data were checked by AT.

**Funding** This work was supported by the Economic and Social Research Council (Grant number ES/J00215/1).

**Competing interests** None declared.

**Patient consent** This paper does not contain personally identifiable information regarding any participant.

**Ethics approval** University of Sheffield Psychology Department Ethics Committee.

**Provenance and peer review** Not commissioned; externally peer reviewed.

**Data sharing statement** The protocol and data files are available from the corresponding author on request.

