## [Reviewer comments · BMJ Open]

ARTICLE DETAILS

TITLE (PROVISIONAL)	A mixed methods survey of social anxiety, anxiety, depression, and wig use in alopecia.
AUTHORS	Montgomery, Kerry; White, Caroline; Thompson, Andrew

VERSION 1 - REVIEW

REVIEWER	Franziska Matzer Department of Medical Psychology and Psychotherapy, Medical University of Graz, Graz, Austria
REVIEW RETURNED	12-Jan-2017

GENERAL COMMENTS	Although the manuscript addresses an interesting study question (relations between wig use and psychological distress as well as qualitative findings concerning experiences with wigs), it seems not consistent enough to be published at present. Abstract: The way in which the term „alopecia areata“ is used is misleading: it is stated that 33.7% reported a diagnosis of alopecia areata, later in the manuscript it becomes clear that another 31.4% had alopecia universalis and 17.5% had alopecia totalis, which are all subtypes of alopecia areata. Please specify what you mean by this term, probably patchy alopecia? Introduction: Line 25, martial problems – probably you mean marital problems Line 48, NHS: please explain abbreviations the first time they appear in the text Please specify what you mean by “psychological distress” in the objectives Methods The methods section should be re-ordered: descriptions of measures should appear earlier in the text before the analysis section, and information on participants could even be placed in the beginning of the results section. Line 33: delete “,” Analysis: please specify statistical methodology (program used for calculations, type of t-test, p-values that were considered significant). Participants: It would be interesting if authors could estimate on the response rate. Although this won't be possible concerning social media, it would be interesting to know how many people the mailing list included. This topic should also be included in the limitations. For better overview it would be good to have all demographic data, including all 11 types of alopecia, in a table. It is unclear if all respondents used wigs, please indicate to which sample number the percentages given refer to. In addition, information on missing data is missing. Did you receive
---

	338 complete data sets? Measures: Please include a list of all survey questions used (either in the text or listed in a table). This section should also have a separate heading. Results: Line 54: insert percentage Table 1: please include sample size Page 12, line 21: I don't understand how participants felt more self-conscious and less confident wearing wigs Discussion: In general the discussion should be more focused on the study questions. Please state how your findings refer to the existing literature, e.g. the levels of psychological distress you found and any relations to wig use. Also the finding that age was negatively correlated with anxiety needs discussion and should be related to literature. These findings refer to 2 of the 3 study questions and should be discussed in more detail. Authors suggest that improvements in wig quality would increase confidence in social interaction. I have the impression that this conclusion cannot be drawn from the data given – here other factors that are discussed later such as poor self-consciousness or pre-occupation with appearance must be taken into account, as wigs per don't seem to reduce psychological distress in this patient group.
--	--

REVIEWER	Lucia Zannini University of Milan, Italy
REVIEW RETURNED	24-Jan-2017

GENERAL COMMENTS	Thank you for giving me the opportunity to review this paper. This is an interesting study on an option, wearing a wig when suffering from alopecia, which has high impact on perceived quality of life in large groups of patients. This paper is clear and the integration between quantitative and qualitative data contribute to a deeper understanding of the experience of living with alopecia and the patients' needs that derive from that experience. I have some suggestions that Authors can consider:  - In the Abstract, I would eliminate the phrase: "The provision of psychological interventions..." (p. 3, lines3-5), since it is not clear from where this conclusion derives from. - In the Introduction, I think that a more complex and multi-faced perspective on the decision of wearing a wig should be given. For instance, Rosman (2004) has observed that using camouflage is most common in those patients who see chemotherapy as a harmful and destructive treatment. Other patients, in contrast, treat hair loss as commonplace: in this case, wearing a wig is played down and banalised. These results come from studies regarding patients who, in the majority of the cases, live baldness as a transitory experience. Even if your study does not consider patients suffering from chemotherapy induced alopecia, I think that you should make some reflections on how wearing a wig becomes a common need, when alopecia is not a transitory condition. - Regarding ethical aspects, I think that should be specified if, for patients < 18 informed consent was obtained by parents. - In the section "Measures" you should report which r results are
---

	obtained when the Social phobia inventory, the Patient health questionnaire and the Generalized anxiety disorder questionnaire are administered to “normal” population (not suffering from alopecia).  - Literature states that alopecia perception and the process of stigmatization vary a lot, according to patients’ culture. Even if your sample is composed by patients theoretically belonging to the same culture, I think that some differences could be found between people living in large towns or in rural areas. Accordingly, results can be exposed also in light of this variable. - In the Discussion, you conclude, “wigs appear to be much more than a ‘cosmetic’ tool”. Some reflections on the process of accompanying people in choosing to wear a wig, giving support and advice, may be included here. Rosman, S. (2004), Cancer and stigma: experience of patients with chemotherapy-induced alopecia. Patient Education and Counseling, 52(3), 333-339.
--	---

VERSION 1 – AUTHOR RESPONSE

Reviewers comments

Although the manuscript addresses an interesting study question (relations between wig use and psychological distress as well as qualitative findings concerning experiences with wigs), it seems not consistent enough to be published at present.

Abstract:

The way in which the term „alopecia areata“ is used is misleading: it is stated that 33.7% reported a diagnosis of alopecia areata, later in the manuscript it becomes clear that another 31.4% had alopecia universalis and 17.5% had alopecia totalis, which are all subtypes of alopecia areata. Please specify what you mean by this term, probably patchy alopecia?

Thank you for this comment. It is unclear that we are referring to them as subtypes therefore we have grouped alopecia areata and its subtypes (82.6%) in the abstract and throughout the paper

Introduction:

Line 25, marital problems – probably you mean marital problems

Thank you for bringing this to our attention. We have corrected this.

Line 48, NHS: please explain abbreviations the first time they appear in the text

Thank you for bringing this to our attention. This has been changed in the bullet points and the text.

Please specify what you mean by “psychological distress” in the objectives

We have changed this to be specific to what we are measuring (social anxiety, anxiety and depression) throughout the text.

Methods

The methods section should be re-ordered: descriptions of measures should appear earlier in the text before the analysis section, and information on participants could even be placed in the beginning of the results section.

Thank you for the comment on the methods section. We have moved the description on measures to earlier in the methods section before the analysis. We have also moved participant characteristics to the results section as suggested by the reviewer.

Line 33: delete “;”

This has now been deleted, thank you for bringing this to our attention.

Analysis: please specify statistical methodology (program used for calculations, type of t-test, p-values that were considered significant).

Thank you to the reviewer for this comment. We have added the programme used for statistical analysis (SPSS version 22), the type of test (independent sample t test and Pearson correlation). We have also added that we used an alpha level of .05 for all statistical tests.

Participants:

It would be interesting if authors could estimate on the response rate. Although this won't be possible concerning social media, it would be interesting to know how many people the mailing list included. This topic should also be included in the limitations.

We are unable to estimate a response rate. Whilst we could provide the number of people on the mailing list the study was also advertised on social media and therefore it is unclear how many participants were recruited via the mailing list or social media. We have included this as a limitation as suggested by the reviewer.

For better overview it would be good to have all demographic data, including all 11 types of alopecia, in a table.

We have provided an overview of demographic data, including type of alopecia, gender and ethnicity (Table 1) Age is referred to in the text.

It is unclear if all respondents used wigs, please indicate to which sample number the percentages given refer to.

Thank you for this comment. We have included a table which gives details on the percentage of participants wearing wigs and the frequency participants chose to wear wigs (Table 2).

In addition, information on missing data is missing. Did you receive 338 complete data sets?

The total number of responses for the quantitative data has been included in the tables and text (N = 338). The number of people responding to free text questions has been indicated in the results. We hope this addresses the reviewers comment.

Measures:

Please include a list of all survey questions used (either in the text or listed in a table). This section should also have a separate heading.

We have now included a list of survey questions in the methods section.

Results:

Line 54: insert percentage

We have added percentages as requested.

Table 1: please include sample size

We have indicated the sample size in the table title.

Page 12, line 21: I don't understand how participants felt more self-conscious and less confident wearing wigs

We have now amended this text to show the reasons for why participants reported feeling more self-conscious, which included fears of the wig being noticed by others.

Discussion:

In general the discussion should be more focused on the study questions.

We have referred to the study questions throughout the discussion to address the reviewers comment.

Please state how your findings refer to the existing literature, e.g. the levels of psychological distress you found and any relations to wig use. Also the finding that age was negatively correlated with anxiety needs discussion and should be related to literature.

We have addressed this comment by acknowledging levels of anxiety, depression and social anxiety reported in previous studies, and how the current study is consistent with this. The relation between wig usage and psychological distress has not previously been investigated. We have focused on the relationship between psychosocial distress and wigs in the revised submission.

These findings refer to 2 of the 3 study questions and should be discussed in more detail.

We have made reference in the discussion of the main aims of the study to highlight how the findings relate to the aims of the study.

Authors suggest that improvements in wig quality would increase confidence in social interaction. I

have the impression that this conclusion cannot be drawn from the data given – here other factors that are discussed later such as poor self-consciousness or pre-occupation with appearance must be taken into account, as wigs per don't seem to reduce psychological distress in this patient group. We have deleted this comment, in order to, as the reviewer suggests, focus on the psychological consequences of alopecia.

Reviewer 2

Thank you for giving me the opportunity to review this paper. This is an interesting study on an option, wearing a wig when suffering from alopecia, which has high impact on perceived quality of life in large groups of patients. This paper is clear and the integration between quantitative and qualitative data contribute to a deeper understanding of the experience of living with alopecia and the patients' needs that derive from that experience. I have some suggestions that Authors can consider: -

In the Abstract, I would eliminate the phrase: "The provision of psychological interventions..." (p. 3, lines3-5), since it is not clear from where this conclusion derives from. –

Removed as per reviewer request.

In the Introduction, I think that a more complex and multi-faced perspective on the decision of wearing a wig should be given. For instance, Rosman (2004) has observed that using camouflage is most common in those patients who see chemotherapy as a harmful and destructive treatment. Other patients, in contrast, treat hair loss as commonplace: in this case, wearing a wig is played down and banalised. These results come from studies regarding patients who, in the majority of the cases, live baldness as a transitory experience. Even if your study does not consider patients suffering from chemotherapy induced alopecia, I think that you should make some reflections on how wearing a wig becomes a common need, when alopecia is not a transitory condition. –

We have added in some further consideration of the psychological variables that might contribute to wig use. As the reviewer highlights, the study does not focus on chemotherapy induced alopecia and therefore we have focused on dermatological studies to address this point. In order to address the reviewers comments we have discussed the impact of living with an altered appearance, drawing on the concept of stigmatisation.

Regarding ethical aspects, I think that should be specified if, for patients < 18 informed consent was obtained by parents. –

We have added a line in the measures section outlining that participants under 16 were advised to complete with a parent or guardian.

In the section "Measures" you should report which results are obtained when the Social phobia inventory, the Patient health questionnaire and the Generalized anxiety disorder questionnaire are administered to "normal" population (not suffering from alopecia). –

We have added in the results of general population and mental health samples on each psychometric questionnaire.

Literature states that alopecia perception and the process of stigmatization vary a lot, according to patients' culture. Even if your sample is composed by patients theoretically belonging to the same culture, I think that some differences could be found between people living in large towns or in rural areas. Accordingly, results can be exposed also in light of this variable. –

This is an interesting reflection; however, we are unable to analyse this as we do not have information on participants geographical location.

In the Discussion, you conclude, "wigs appear to be much more than a 'cosmetic' tool". Some reflections on the process of accompanying people in choosing to wear a wig, giving support and advice, may be included here.

We acknowledge the reviewers comment, this is a very important aspect of care for alopecia patients; however, this was not discussed by participants, and therefore the conclusion focuses on the psychological aspects of support which may be beneficial.

VERSION 2 – REVIEW

REVIEWER	Franziska Matzer Department of Medical Psychology and Psychotherapy, Medical University of Graz, Austria
REVIEW RETURNED	10-Mar-2017

GENERAL COMMENTS	Thank you for the possibility to review the revised version of this manuscript. The authors have given detailed answers to all reviewers' comments and have addressed their suggestions throughout the manuscript. I think that the paper has improved in clarity and consistency and can be published.
---

REVIEWER	Lucia Zannini Department of Biomedical Sciences, University of Milan, Italy
REVIEW RETURNED	19-Mar-2017

GENERAL COMMENTS	Thank you for giving me the opportunity to review again this paper. I think that in this new version your paper is clearer. Nevertheless, when I read again the paper and went back to the abstract, I noticed some incongruences: - In the "objectives" section you should specify that the type of alopecia you consider is just alopecia deriving from a dermatological condition. Furthermore, you declare that the paper is aimed at reporting financial implications of wearing wigs. I think this point is not adequately developed in the paper, so I would change the last phrase in "The study also sought to report on experiences of wearing wigs in social situations and the relationship between this camouflage and social confidence".- Accordingly, in the "discussion" section, I would underline that less than half of the participants reported that wearing a wig had a positive impact on their everyday life. For this reason, I would change the last phrase of the Discussion as follows: "Overall, 46% of participants reported that wearing a wig had a positive impact on their everyday life. Negative experience of wearing a wig are related to its cost and fears of wig being noticed. Therefore, financial as well as psychological interventions would be beneficial for people with alopecia, alongside wig provision". Consistent with your results, it seems that wearing a wig is not "the" solution to cope with alopecia, but it is seen as "the lesser of two evils". As it emerges from other studies, wig provision should be characterized by (financial) assistance to get a good quality wig, as well as psychological support to people who are facing alopecia. Wearing a wig is not the solution of the patients' difficulties, but one of the strategies that can be included in the more global process of caring for those patients.
--

VERSION 2 – AUTHOR RESPONSE

Reviewer: 1

Franziska Matzer

Department of Medical Psychology and Psychotherapy,
Medical University of Graz, Austria

Please state any competing interests or state 'None declared': None declared

Please leave your comments for the authors below

Thank you for the possibility to review the revised version of this manuscript. The authors have given detailed answers to all reviewers' comments and have addressed their suggestions throughout the manuscript. I think that the paper has improved in clarity and consistency and can be published.

Reviewer: 2

Lucia Zannini

Department of Biomedical Sciences, University of Milan, Italy

Please state any competing interests or state 'None declared': None declared

Please leave your comments for the authors below

Thank you for giving me the opportunity to review again this paper.

I think that in this new version your paper is clearer.

Nevertheless, when I read again the paper and went back to the abstract, I noticed some incongruences:

- In the "objectives" section you should specify that the type of alopecia you consider is just alopecia deriving from a dermatological condition. Furthermore, you declare that the paper is aimed at reporting financial implications of wearing wigs. I think this point is not adequately developed in the paper, so I would change the last phrase in "The study also sought to report on experiences of wearing wigs in social situations and the relationship between this camouflage and social confidence".

Thank you for this comment, we have amended the text according. In the text we have stated 'The study also sought to report on experiences of wearing wigs in social situations and the relationship with social confidence'

- Accordingly, in the "discussion" section, I would underline that less than half of the participants reported that wearing a wig had a positive impact on their everyday life. For this reason, I would change the last phrase of the Discussion as follows: "Overall, 46% of participants reported that wearing a wig had a positive impact on their everyday life. Negative experience of wearing a wig are related to its cost and fears of wig being noticed. Therefore, financial as well as psychological interventions would be beneficial for people with alopecia, alongside wig provision".

Thank you for highlighting the discussion section of the abstract, we agree that the sentence proposed by the reviewer does sit more consistently with the findings. We have amended the text. We hope that our proposal that wig provision should be provided is clear that this means people receive support with wig provision (which includes financial).

Consistent with your results, it seems that wearing a wig is not "the" solution to cope with alopecia, but it is seen as "the lesser of two evils". As it emerges from other studies, wig provision should be characterized by (financial) assistance to get a good quality wig, as well as psychological support to people who are facing alopecia. Wearing a wig is not the solution of the patients' difficulties, but one of the strategies that can be included in the more global process of caring for those patients.

Thank you for your comment and we hope the paper provides support to this notion that people living with alopecia would benefit from further support.